# Generation of targeted homozygosity in the genome of human induced pluripotent stem cells

Yasuhide Yoshimura[1]*, Ayako Yamanishi[1], Tomo Kamitani[1], Jin-Soo Kim[2], Junji Takeda[1¤]*

**1** Department of Genome Biology, Graduate School of Medicine, Osaka University, Suita, Osaka, Japan,
**2** Center for Genome Engineering, Institute for Basic Science, Seoul, South Korea

¤ Current address: Research Institute for Microbial Diseases, Osaka University, Suita, Osaka, Japan
* jjtakeda@biken.osaka-u.ac.jp (JT); yoshimura_y@gts.med.osaka-u.ac.jp (YY)

**Data Availability Statement:** All relevant data are within the article and its supporting information files.

## Abstract

When loss of heterozygosity (LOH) is correlated with loss or gain of a disease phenotype, it is often necessary to identify which gene or genes are involved. Here, we developed a region-specific LOH-inducing system based on mitotic crossover in human induced pluripotent stem cells (hiPSCs). We first tested our system on chromosome 19. To detect homozygous clones generated by LOH, a positive selection cassette was inserted at the AASV1 locus of chromosome 19. LOHs were generated by the combination of allele-specific double-stranded DNA breaks introduced by CRISPR/Cas9 and suppression of Bloom syndrome (*BLM*) gene expression by the Tet-Off system. The BLM protein inhibitor ML216 exhibited a similar crossover efficiency and distribution of crossover sites. We next applied this system to the short arm of chromosome 6, where human leukocyte antigen (HLA) loci are located. Genotyping and flow cytometric analysis demonstrated that LOHs associated with chromosomal crossover occurred at the expected positions. Although careful examination of HLA-homozygous hiPSCs generated from parental cells is needed for cancer predisposition and effectiveness of differentiation, they may help to mitigate the current shortcoming of hiPSC-based transplantation related to the immunological differences between the donor and host.

## Introduction

We and others have previously reported a method to induce loss of heterozygosity (LOH) in mouse embryonic stem cells (ESCs) under suppression of the Bloom syndrome (*Blm*) gene [1] [2] [3]. Suppression of *Blm* enhanced the rate of mitotic recombination, and crossovers occurred randomly across the observed chromosome [4]. When DNA double-stranded breaks (DSBs) were introduced by meganuclease I-SceI under *Blm* suppression, the rate of LOH was further enhanced by crossovers at sites close to the allele-specific I-SceI-cutting site [4]. We analyzed mouse ESC clones after generating the crossovers and found that LOH occurred across entire genomic regions from crossover points to telomeres.

**Funding:** This work was supported by the Japan Society for the Promotion of Science (KAKENHI grant numbers 24241064 and 15K15420). The funders had no role in study design, data collection and analysis, decision to publish, or preparation of the manuscript.

**Competing interests:** The authors have declared that no competing interests exist.

To date, a system that induces LOH in a targeted region on a specific chromosome in cells, such as human induced pluripotent stem cells (hiPSCs), has not been established yet. We hypothesized that a feasible method to generate region-specific LOH in hiPSC lines would be as follows. Assuming that parental hiPSCs bear a heterozygous mutation on the target chromosome and an allele-specific DSB is introduced on that chromosome during the 4N stage of the cell cycle, the genomic region from the site of the DSB to the telomere would contain either a homozygous mutation or no mutation after crossover and chromosome segregation (Fig 1A). This phenomenon can then be used to identify genes responsible for certain diseases. When the mutation "X" is a dominant mutation (parental hiPSCs have the disease phenotype), some of the clones do not have the disease phenotype after crossover (case #b in Fig 1A). However, when X is a recessive mutation (parental hiPSCs do not have the disease phenotype), some of the clones have the disease phenotype after crossover (case #a in Fig 1A).

This study revealed that both allele-specific DSBs mediated by CRISPR/Cas9 and *BLM* suppression allow targeted homozygosity in hiPSCs. Therefore, this system is applicable to in vitro genetic analysis of hiPSCs, when crossbreeding experiments are not possible.

## Methods

### Vector construction

*BLM*-targeting vector: To prepare the human *BLM*-targeting vector, we changed the *tk* promoter to the *hPGK* promoter and modified the Kozak sequence for tTA translation from AGGATT to GCCACC in the mouse *Blm*-targeting vector [1]. We also removed the artificial intron sequence from the mouse *Blm*-targeting vector [1]. The left and right arms of the targeting vector were prepared by PCR using primers hBLMLarmF1 and hBLMLarmR1, and hBLMRarmF1 and hBLMRarmR3, respectively (S1 Table). Chromosome 19 *AAVS1*-vector construction with the cNP cassette: The cNP cassette used in this study was derived from the cNP cassette used in mouse ESCs [2] with the following three modifications. The *Pgk* promoter was changed to the *hEF1α promoter*, the *Δtk g*ene was removed, and *Lox2272*s were replaced by *loxP*s. The left and right arms of the targeting vector were prepared by PCR using primers AAVS1-LF1 and AAVS1-LR1, and AAVS1-RF1 and AAVS1-RR1, respectively (S1 Table). Chromosome 6 HLA region-targeting vector with cNP in PB: The insulator sequence (TGCTTGTCCTTCCTTCCTGTAACACAGCCATTAAACCAGGAGCATCGCCCTTCCCCGGCCCT CAGGTAAGAGGACCAAATACCGTAGCCGTTTCCAATTTCAGTCCTTTAGCGCCACCTGGTGC TAACTACTCTATCACGCTTTTATCCAATAACTACCTTTGTAAATTTCCTTTCAAAAGTTCTG GCCGGGCGCGGTGGCTCACGCTTGTAATCCCAGCACTTTGTGAGGGGTCAGGAGTTC; chromosome 7, bp 39519983–39520225: GRCh38) [5] was flanked by the cNP cassette. The insulator and cNP cassette were inserted into the piggyBac transposon vector [6], resulting in the cNP in PB. The left and right arms of the targeting vector were prepared by PCR using primers chr6-Asc1-up and chr6-Asc1-low, and chr6-Pac1-up and chr6-Pac1-low, respectively (S1 Table). The iCre gene was expressed under the control of the EF1α promoter.

### Cell culture

hiPSC lines [7] were grown in hESC serum-free human ESC (hESC) medium consisting of DMEM/F-12 (Life Technologies) supplemented with 20% knockout serum replacement (Life Technologies), 2 mM L-glutamine, 1× nonessential amino acids (Life Technologies), 0.1 mM 2-mercaptoethanol, and 5 ng/mL basic fibroblast growth factor (Katayama Chemical Industries) on Synthemax II-SC-coated tissue culture dishes (Corning). The cells were passaged using Accutase (Sigma) and seeded with the Rho kinase inhibitor Y-27632 (10 μM; LC Laboratories).

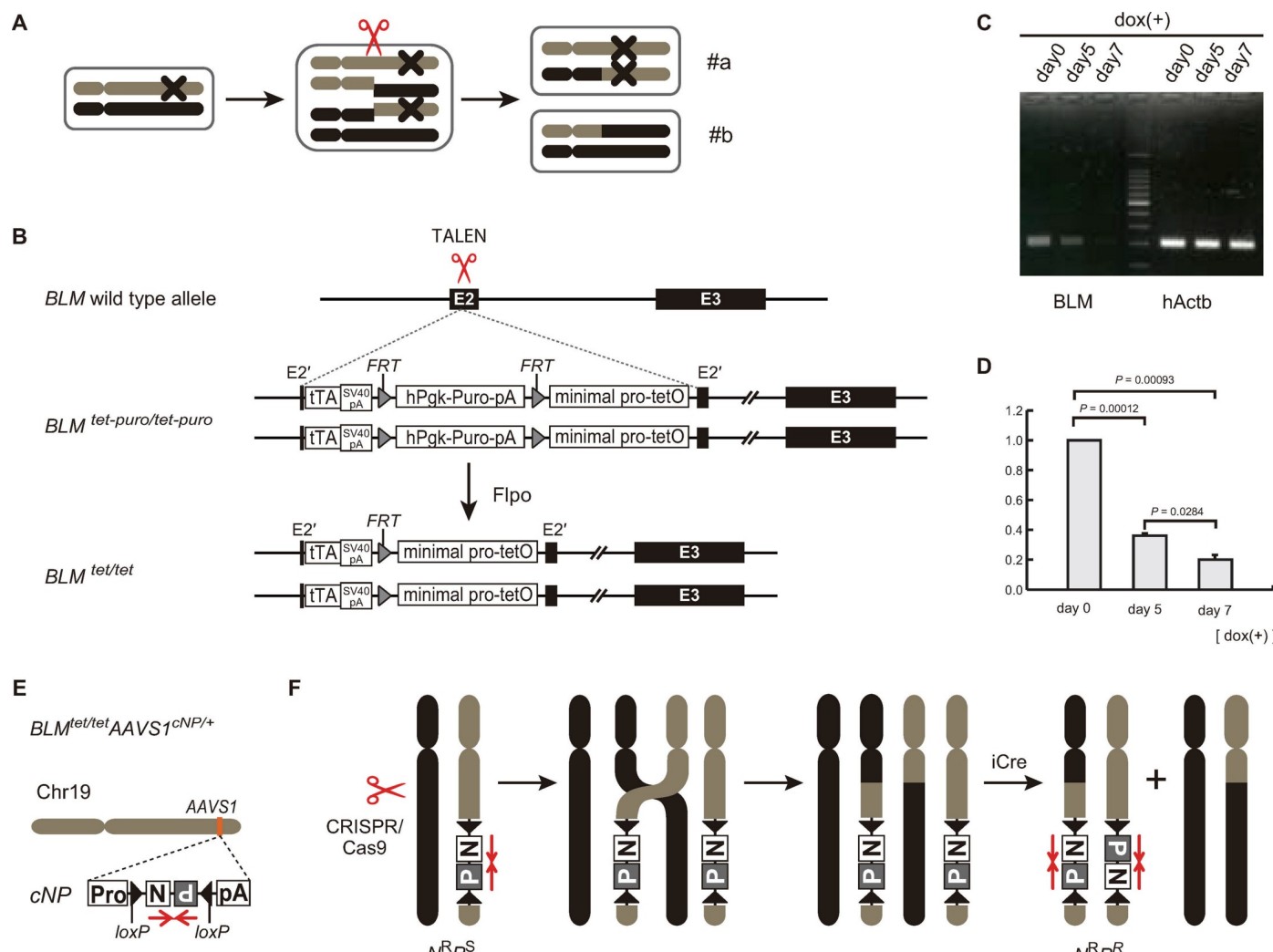

**Fig 1. Establishment of a *BLM* transcript regulation system and detection of crossovers by a reporter cassette in hiPSCs.** (A) Schematic representation of a crossover induced by a double-stranded DNA break during the 4N stage of the cell cycle in hiPSCs. X is a dominant mutation. After segregation, case #a cells have the disease phenotype, while case #b cells are normal. (B) Targeting of the Tet-Off cassette to both alleles of the *BLM* locus with the assistance of TALEN (*BLM*^tet-puro/tet-puro^). The selection cassettes were removed by Flippase (Flpo), the hiPSC-*BLM*^tet/tet^ cells were established. (C) Electrophoresis of qRT-PCR products. Total RNA from $2 \times 10^5$ hiPSC-*BLM*^tet/tet^ cells was collected at days 0, 5, and 7 after dox treatment, and qRT-PCR was performed. (D) Histogram data represent the mean ± SEM of three independent experiments. Comparison of two groups with normally distributed variables was performed using the Student's *t*-test by Kaleida Graph software. (E) Targeting of the double selection cassette (*cNP*) to the AAVS1 locus (hiPSC-*BLM*^tet/tet^*AAVS*^cNP/+^). (F) Schematic representation of mitotic crossover during the 4N stage. DSBs were introduced by CRISPR/Cas9.

## Targeting

To target the *BLM* locus, hiPSCs were transfected as a single cell suspension by electroporation (Neon Transfection System; Invitrogen) using $1 \times 10^6$ cells in a 100 μL tip with 8 μg total DNA (TALEN left, 2 μg; TALEN right, 2 μg; *BLM*-targeting vector, 4 μg). TALEN sequences are shown in S2 Table. After one pulse of electroporation at 1,100 V with a 30 ms pulse width, the cells were plated on Synthemax II-SC-coated tissue culture dishes in hESC medium containing 10 μM Y-27632 without puromycin. Selection was initiated with 0.5 μg/mL puromycin at 2 days after transfection. The targeted clones were first selected by PCR with primers hBLM-3 and tTA2, and then the targeted biallelic clones (*BLM*^tet/tet^ hiPSCs) were chosen by PCR with primers hBLM-1, hBLM-2, and tTA-1. Then, the selection cassettes were removed by Flippase

(Flpo), and the hiPSC-*BLM^{tet/tet}* cells were thereby established (S1 Table). To target the *AAVS1* locus, *BLM^{tet/tet}* hiPSCs were transfected as a single cell suspension by electroporation using $1 \times 10^6$ cells in a 100 μL tip with 8 μg total DNA (*AAVS1-ZFN* left, 2 μg; *AAVS1-ZFN* right, 2 μg; *AAVS1*-targeting vector, 4 μg). The AAVS1-ZFN sequences have been described previously [8]. After one pulse of electroporation at 1,200 V with a 30 ms pulse width, the cells were plated on Synthemax II-SC-coated tissue culture dishes in hESC medium containing 10 μM Y-27632 without G418. Selection was initiated with 100 μg/mL G418 at 2 days after transfection. The targeted clones were first selected by PCR with primers AAVS1-LF2 and Puro-L2, and then the targeted monoallelic clones (*BLM^{tet/tet} AAVS1^{cNP/+}*hiPSCs) were chosen by PCR with primers AAVS1-C-F, AAVS1-C-R, and Puro-L2 (S1 Table). To target the telomeric region of the short arm of chromosome 6, hiPSCs were transfected as a single cell suspension by electroporation using $1 \times 10^6$ cells in a 100 μL tip with 8 μg pX330 (AGAAAATAGCCGCCACTTAA as the gRNA) and 3.5 μg HLA-targeting vector with cNP in PB. After one pulse of electroporation at 1,200 V with a 30 ms pulse width, the cells were plated on Synthemax II-SC-coated tissue culture dishes in hESC medium containing 10 μM Y-27632. Selection was initiated with 100 μg/mL G418 at 2 days after transfection. The concentration of G418 was increased to 150 μg/mL at 4 days post-transfection. The targeted clones were first selected by PCR with primers chr6-TG-up and hEF1a-TG, and then the targeted monoallelic clones (*telHLA^{cNP/+}*hiPSCs) were chosen by PCR with the primers chr6-TG-up, chr6-TG-low, and hEF1a-TG (S1 Table).

### Introduction of allele-specific DSBs in chromosome 19

*BLM^{tet/tet}AAVS1^{cNP/+}* hiPSCs were transfected as a single cell suspension by electroporation using $1 \times 10^6$ cells in a 100 μL tip with 9.5–11 μg total DNA (6.5–8.0 μg pX330 containing various gRNA sequences and 3 μg iCre recombinase plasmid). *BLM^{tet/tet}AAVS1^{cNP/+}* hiPSCs were pretreated with dox (1 μg/mL) from 7 days before transfection. The gRNA sequences are listed in S3 Table. After one pulse of electroporation at 1,300 V with a 20 ms pulse width, the cells were plated on Synthemax II-SC-coated tissue culture dishes in hESC medium containing 10 μM Y-27632 without puromycin or G418. At 2 days after transfection, selection was initiated with 0.5 μg/mL puromycin. At 4 days after transfection, the selection condition was changed to 1 μg/mL puromycin and 100 μg/mL G418. At 6 days after selection, the G418 concentration was increased to 200 μg/mL G418.

### Introduction of allele-specific DSBs in chromosome 6

*telHLA^{cNP/+}* hiPSCs were transfected using the same protocol for chromosome 19. The gRNA sequences are listed in S3 Table. ML216 (12.5 μM) was applied from 12 h before transfection until 2 days after transfection. At 2 days post-transfection, selection was initiated with 0.2 μg/mL puromycin. At 4 days post-transfection, the selection condition was changed to 0.25 μg/mL puromycin and 25 μg/mL G418. At 6 days after selection, the selection condition was changed again to 0.33 μg/mL puromycin and 50 μg/mL G418.

### RNA extraction and quantitative real-time polymerase chain reaction (qRT-PCR)

Total RNA was extracted from cells with an RNeasy Plus Micro Kit (Qiagen). First-strand cDNA was synthesized from 800 ng total RNA with random hexamer primers using Super-Script III (Invitrogen) at 50°C for 60 min. qRT-PCR was performed to measure *BLM* gene expression. Total RNA was reverse transcribed to cDNA using the SuperScript III First-Strand Synthesis System for qRT-PCR (Invitrogen). qRT-PCR was performed with the Applied

Biosystems 7900HT Fast Real-Time PCR System using the following PCR primer sets: RTPCR-leader-F and RTPCR-exon3-R, hActb-RTPCR-F, and hActb-RTPCR-R (S1 Table).

## Quantification of pluripotency marker gene expression

Total RNA was isolated from parental fibroblasts, hiPSC-*telHLA$^{cNP/+}$*, and hiPSC-*telHLA$^{cNP/+}$*ML216(+)HLA I-III-CRISPR(+). Library preparation was performed using a TruSeq stranded mRNA sample prep kit (Illumina, San Diego, CA), according to the manufacturer's instructions. Whole transcriptome sequencing was applied to the RNA samples through Illumina HiSeq 2500 and 3000 platforms in the 75-base single-end mode. Illumina Casava ver.1.8.2 software was used for base calling. The sequenced reads were mapped to human reference genome sequences (hg19) using TopHat ver. 2.0.13 in combination with Bowtie2 ver. 2.2.3 and SAMtools ver. 0.1.19. The number of fragments per kilobase of exon per million mapped fragments (FPKM) was calculated using Cufflinks ver. 2.2.1. The FPKM values were calculated from the respective sequence data, and the analyses were performed using iDEP85 (http://bioinformatics.sdstate.edu/idep/). The SNP-array data have been deposited in the Genomic Expression Omnibus under accession number (GSE134441,GSE137657, GSE138093).

## Screening for positive clones with crossover occurring in chromosomes 19 and 6

DNA was extracted from hiPSCs with a DNeasy Blood & Tissue Kit (Qiagen). Competitive PCR was performed to detect positive clones of chromosome 19 using primers AAVS1-C-F, AAVS1-C-R, and Puro-L2 (S1 Table). Cycling conditions were initial denaturation at 94˚C for 2 min, followed by 35 cycles of 98˚C for 10 s, 68˚C for 90 s, and 72˚C for 7 min. The wildtype allele was detected at 275 bp, the neomycin-selective allele was detected at 407 bp, and the puromycin-selective allele was detected at 1,435 bp (S1 Fig). We performed the same screening for positive clones of chromosome 6 using primers chr6-TG-up, chr6-C-low, and Puro-L2 (S1 Table). The wildtype allele was detected at 924 bp, the neomycin-selective allele was detected at 1,355 bp, and the puromycin-selective allele was detected at 2,264 bp (data not shown).

## Determination of crossover points in chromosomes 19 and 6

First, we identified heterozygous SNPs by referring to the SNP database dbSNP https://www.ncbi.nlm.nih.gov/snp (S4 Table). After identifying positive clones by competitive PCR, we extracted the genomic DNA and analyzed the crossover points (S4 Table). We defined the crossover points as the positions equidistant from the heterozygous and homozygous SNPs.

## HLA genotyping

First, target DNA was amplified by PCR using group-specific primers LABType SSO HLA A Locus, LABType SSO HLA B Locus, LABType SSO HLA C Locus, and LABType SSO HLA DRB1. The PCR products were biotinylated, allowing detection with R-phycoerythrin-conjugated streptavidin. The PCR products were denatured and allowed to rehybridize to cDNA probes conjugated with fluorescently coded microspheres (One Lambda). A LABScan 100 flow analyzer (One Lambda) identified the fluorescence intensity of phycoerythrin on each microsphere. HLA alleles or allele groups were determined in the sample by matching the pattern of positive and negative bead IDs with the information in the LABType SSO worksheet or using HLA Fusion 4.1 HotFix 2 (both from One Lambda).

## Flow cytometry

hiPSC suspensions were washed with phosphate-buffered saline containing 1% (w/v) bovine serum albumin and 0.05% (v/v) sodium azide and then incubated with the appropriate antibodies for 30 min in the dark on ice. Samples were analyzed on a BD FACSCanto II (BD Biosciences). Data were analyzed with FlowJo software (Tomy Digital Biology). The antibodies used for flow cytometry were a FITC-conjugated anti-HLA-A2 antibody (343303; BioLegend), anti-HLA-A32/25 monoclonal IgM antibody (0136HA; One Lambda), and Alexa Fluor 647-conjugated goat anti-mouse IgM (ab150123; Abcam).

## SNP array analysis

To analyze the SNP copy number, genomic DNA was isolated from parental fibroblast, hiPSCs, hiPSC-$BLM^{tet/tet}AAVS1^{cNP/+}$, hiPSC-$BLM^{tet/tet}AAVS1^{cNP/+}$–Dox(+)9M-CRISPR(+),–Dox(+)14M-CRISPR(+),–Dox(+)19M-I-CRISPR(+),–ML216(+)9M-CRISPR(+),–ML216(+) 14M-CRISPR(+),–ML216(+)19M-I-CRISPR(+), and hiPSC-$telHLA^{cNP/+}$ML216(+)HLA I-III--CRISPR(+) and hybridized to an Infinium Omni5-4 v1.2 BeadChip (Illumina). Data were analyzed using GenomeStudio (Illumina). The SNP array data have been deposited in the Genomic Expression Archive under accession number E-GEAD-283.

## Results

### Establishment of a *BLM* transcript regulation system and detection of crossovers by a reporter cassette in hiPSCs

A Tet-Off cassette was used for transient suppression of *BLM* expression. The Tet-Off cassette used for mouse ESCs [1] was modified and inserted into both alleles of the *BLM* locus with the assistance of transcription activator-like effector nucleases (TALENs) followed by removal of the selection cassette (S1 Fig), resulting in the BLM^tet/tet allele (Fig 1B). The level of BLM transcripts in BLM^tet/tet hiPSCs without the addition of doxycycline (dox) was equivalent to that observed in parental hiPSCs (S2 Fig) The BLM expression level was then checked at 7 days after the addition of 1 μg/mL dox, and BLM transcripts were measured by quantitative PCR (qPCR). The transcripts in BLM^tet/tet hiPSCs were downregulated to 20% of that in the control (Fig 1C and 1D). We also performed 8 days of treatment with 1 μg/mL dox and found that the level of BLM transcripts was approximately 15% of that in the control (S3 Fig), and BLM^tet/tet hiPSCs appeared to be unhealthy with longer exposure to dox. Thus, we chose 7 days of treatment with dox.

To detect crossover events in the hiPSCs, a conditionally convertible selection cassette (cNP) was first inserted at the safe-harbor AAVS1 locus of chromosome 19, resulting in the BLM^tet/tet AAVS1^cNP/+ allele (Fig 1E) [2]. The cNP cassette included neomycin (N) and puromycin (P) resistance genes. The N and P genes were arranged in a tail-to-tail configuration and flanked by inversely oriented loxP sequences. Under the default condition, the promoter in front of N drove expression of N but not P, resulting in G418-resistant and puromycin-sensitive hiPSCs ($N^{R}P^{S}$). Upon codon-improved Cre recombinase (iCre) expression, inversion of the cNP cassette occurred, resulting in loss of G418 resistance and gain of puromycin resistance. We used iCre to enhance Cre-mediated cassette inversion [9]. The iCre-mediated cassette inversion was estimated by surviving colonies after the drug selections (S4 Fig). Since the surviving colonies with puromycin were approximately 20%-30% of those with G418, the efficiency of the inversion was 20%-30%. Next, allele-specific DSBs mediated by the CRISPR/Cas9 system were introduced centromeric to the cNP cassette. If crossover occurred in the chromosome bearing the cNP cassette during the 4N stage of the cell cycle, the cNP cassette would be

duplicated after segregation (Fig 1F). These cassette-duplicated cells were selected by both G418 and puromycin after Cre-mediated recombination (N$^R$P$^R$).

This positive selection approach differs from the negative selection used previously, based on loss of genes encoding herpes simplex virus thymidine kinase (TK) or green fluorescent protein (GFP). Recently, Sadhu et al. reported that negative selection based on loss of the *gfp* gene successfully identified yeast clones bearing LOH following the introduction of allele-specific DSBs with the CRISPR/Cas9 system [10]. LOH is often associated with genomic deletion in mammalian cells, and negative selection cannot exclude clones bearing these deletion events [4]. In fact, Riolobos et al. reported that an HLA-homozygous cell line was derived from an HLA-heterozygous human ESC line by negative selection with TK, but the efficiency of crossover events was unexpectedly low [11]. Although they eventually determined the crossover point, it could not be predicted by their system. In contrast, positive selection is able to exclude deletion events. However, it cannot exclude aberrant cassette duplication due to chromosomal or locus amplifications without crossover [2]. However, such aberrant events can be evaluated by competitive PCR to detect whether the wildtype allele is present (S5 Fig).

## Generation of clones with crossovers in chromosome 19

Preliminary experiments were performed on chromosome 19 where the cNP cassette was inserted at the AAVS1 locus. To introduce allele-specific DSBs, we first searched for SNPs in sequences corresponding to that of the protospacer adjacent motif (PAM) and then designed a single guide RNA (sgRNA) for CRISPR/Cas9. The locations of the allele-specific DSBs were 9, 14, and 19 Mb from the centromere of chromosome 19. The overall procedure used to isolate the positive clones is shown in Fig 2A.

Suppression of *BLM* transcripts by dox started at 7 days before transfection, and selection was performed by gradually increasing the G418 and puromycin concentrations (Fig 2A). Because we wanted to eliminate hiPSCs that did not undergo iCre-mediated inversion, we started puromycin selection before the double selection. First, we compared various conditions, *BLM* suppression only, DSBs at 9 Mb only, and the combination of *BLM* suppression and DSBs at 9 Mb, to acquire positive clones with possible crossover. As shown in Fig 2B, we obtained positive clones only when combining *BLM* suppression and DSBs. These results were strikingly different from the previously reported data. In yeast, positive clones with crossovers were obtained by CRISPR-mediated DSBs without suppression of the *BLM* homolog *Sgs1* [10]. Moreover, whereas we could not obtain positive clones by *BLM* suppression only, Yamanishi et al. reported that *Blm* suppression alone was sufficient to induce crossover in mouse ESCs. These contrasting results between hiPSCs and mouse ESCs may be due to either the differing efficiency of *BLM*/*Blm* suppression, the differing characteristics of the used cell lines (hiPSCs vs. mouse ESCs), or both. Incomplete suppression of *BLM* may also be a reason for the low number of colonies.

We next analyzed crossover points after the introduction of DSBs and *BLM* suppression. The parental hiPSC line had many heterozygous SNPs along each chromosome, and crossover points should have occurred between SNP-heterozygous and -homozygous (LOH) sites. We tentatively defined crossover points equidistant to the SNP-heterozygous and -homozygous (LOH) sites. We detected crossover points close to the DSB sites at the chromosomal level (S6A Fig). As shown by the LOH panel in Fig 2C, more detailed analyses at the 9 Mb position revealed that, when crossover occurred centromeric to the DSBs, the distance between the crossovers and DSBs was relatively small (~20 kb). Note that, because all clones were derived from different dishes of cells, clones #1–#8 were an independent clone. In contrast, when crossover occurred telomeric to the DSBs, this distance was relatively large (~300 kb). Clone

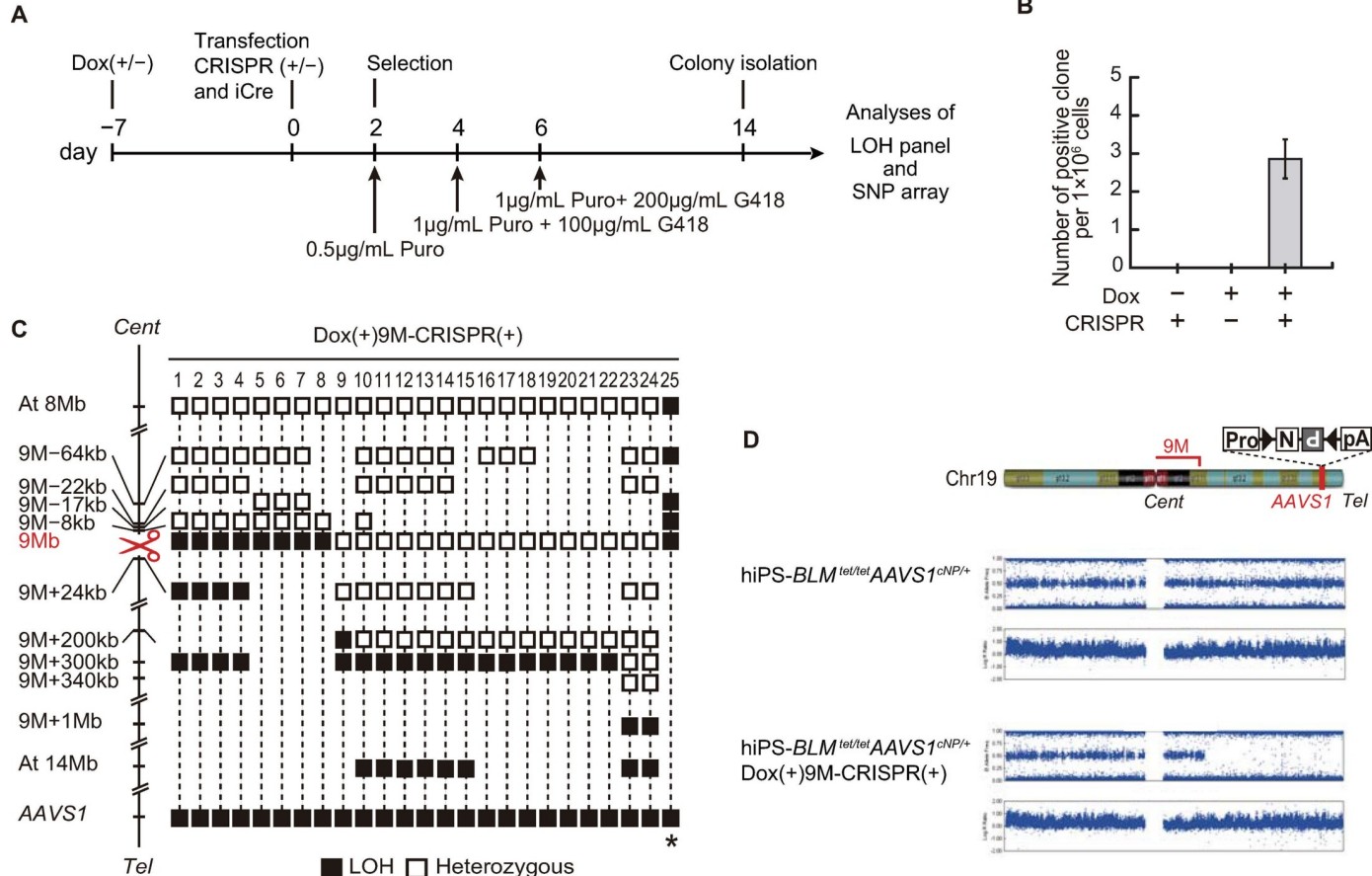

**Fig 2. Generation and analyses of clones with crossovers in chromosome 19.** Procedure for the isolation of $N^R P^R$ clones under the condition of *BLM* transcript suppression. hiPSC-*BLM*$^{tet/tet}$*AAVS1*$^{cNP/+}$ cells were treated with or without dox from 7 days before transfection to the day of transfection. Selection started 2 days after transfection, either with or without the introduction of an allele-specific DSB. (B) Efficiency of crossover under the various conditions at the 9 Mb region of chromosome 19. We picked 439 colonies (eight trials) and performed genotyping (S5 Fig). Bar indicates SEM. (C) Distribution of crossover points at 9 Mb from the centromere of chromosome 19. Twenty-five $N^R P^R$-positive clones were isolated and analyzed using LOH panels and SNP arrays. Open squares indicate the heterozygous genotype. Closed black squares indicate the loss of heterozygosity (LOH) genotype. The DSBs introduced by the CRISPR/Cas9 system are indicated by scissors. (D) A schematic representation of chromosome 19 is shown at the top of the figure. After crossover, SNP array analysis indicated that the genome derived from hiPSC-*BLM*$^{tet/tet}$*AAVS1*$^{cNP/+}$ Dox(+)9M-CRISPR(+) cells showed homozygosity telomeric to DSB and that the cells maintained two copies of the genome.

#25 was exceptional and probably derived from uniparental disomy or a spontaneous DSB close to the centromere, which was independent of the DSB induced by CRISPR/Cas9. Similar distribution patterns were observed at 14 Mb (S6B Fig). To further evaluate this biased distribution pattern, we established two more DSBs (19 Mb-I and 19 Mb-II) close to each other, and the location of 19 Mb-I was centromeric to 19 Mb-II (S6C Fig). Again, both DSBs induced crossovers centromeric to the DSBs. The crossover points induced by 19 Mb-II was not observed for 19 Mb-I. These data implied that the biased distribution pattern was inherent and not influenced by the genome sequence or structure.

## SNP array analysis reveals introduction of homozygosity telomeric to the DSB

The predicted outcome post-crossover in this study would be the same copy numbers (two copies) of the genome and homozygous SNPs telomeric to the DSB. This prediction was

assessed by an SNP array. When the DSB was introduced at the 9 Mb position of chromosome 19, a homozygous SNP pattern was observed telomeric to the DSB on chromosome 19 (9 Mb) (Fig 2D).

## BLM protein inhibitor ML216 has a similar effect on crossovers in chromosome 19

For hiPSC-mediated therapeutics, exogenous genetic materials are undesirable, such as the Tet-Off cassette in the *BLM* gene. We therefore also examined inhibitors of the BLM protein. Nguyen et al. reported that a small molecule inhibitor, ML216, shows strong selectivity for BLM in cultured cells [12]. More recently, Lazzarano et al. showed that ML216 induces mitotic crossover in mouse ESCs. They also reported that transient suppression of the BLM protein did not increase aneuploidy [13]. The procedure used to isolate positive clones generated with the BLM inhibitor ML216 is shown in Fig 3A.

The efficiency of crossover with ML216 was similar to that with *BLM* transcript suppression and dox treatment (Fig 3B). The LOH panel and SNP array analyses showed in a similar pattern of suppression to that induced by the Tet-Off system (Fig 3C and 3D). The effect of ML216 was further evaluated by SNP array analyses at 14 Mb and 19 Mb-I, and a similar crossover pattern was observed (Fig 3E). We performed further SNP analyses with whole chromosomes and we found that the crossover only occurred at the predicted sites (S7 Fig).

## Combination of BLM inhibitor ML216 and DSBs results in homozygosity of HLA class I genes in the short arm of chromosome 6

Because HLA genes are highly polymorphic, most individuals have two HLA haplotypes, namely the HLA-heterozygous state. When established hiPSCs or their differentiated cells, which are most likely HLA-heterozygous, are used for therapeutics as donor cells, the recipient must have the same combination of HLA haplotypes. Our strategy is able to induce HLA homozygosity in the entire region of the HLA locus. Therefore, we can easily choose a much less incompatible combination between the donor of HLA-homozygous hiPSC-derived differentiated cells and recipient.

To generate HLA-homozygous hiPSCs, we inserted the cNP cassette into a piggyBac transposon vector within the telomeric region of the short arm of chromosome 6, where HLA genes are located (hiPSC-*telHLA*$^{cNP/+}$). Then, we introduced an allele-specific DSB between HLA class I and III genes under inhibition of the BLM protein by ML216. The procedure used to isolate the HLA-homozygous clones is shown in Fig 4A, and the location of DSBs introduced by CRISPR/Cas9 and the HLA genotype in parental hiPSCs are schematically presented in Fig 4B.

We obtained two independent positive clones and analyzed the crossover points. Both crossover points were nearby and centromeric to the DSB (Fig 4C, left). The HLA genotypes were consistent with the crossover points (Fig 4C, right). SNP array analysis revealed a homozygous SNP pattern in the short arm of chromosome 6 telomeric to the DSB (Fig 4D), which was consistent with the pattern predicted post-crossover. We analyzed expression of marker genes for pluripotency and found that their expression in hiPS-*telHLA*$^{cNP/+}$ ML216(+) HLA I-III-CRISPR (+) was similar to that in hiPS-*telHLA*$^{cNP/+}$. These data suggested that the clones maintained pluripotency after crossovers (S8 Fig). To confirm the crossovers, we analyzed the surface expression of HLA class I molecules with a haplotype-specific monoclonal antibody against HLA-A (Fig 4E). hiPSC-*telHLA*$^{cNP/+}$ expressed both A32 and A2 haplotypes (S9 Fig). In contrast, hiPSCs after crossover expressed only the A2 haplotype. We therefore successfully

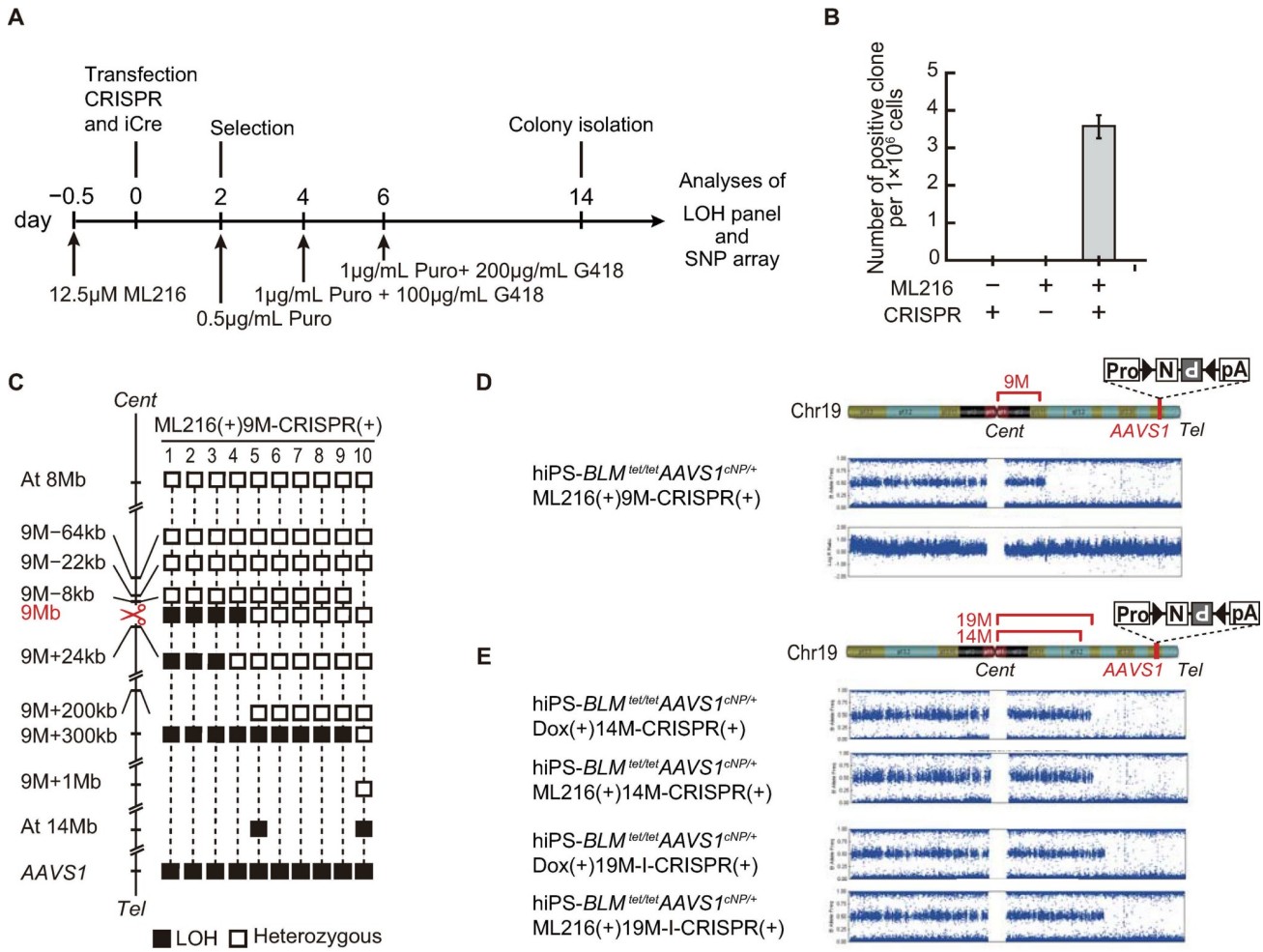

**Fig 3. BLM protein inhibitor ML216 has a similar effect on crossovers in chromosome 19.** Procedure for the isolation of $N^R P^R$ clones under the condition of BLM protein inhibition. hiPSC-$BLM^{tet/tet}AAVS1^{cNP/+}$ cells were treated with ML216 from 12 h before transfection to 1 day after transfection. (B) Efficiency of crossover with ML216 treatment at the 9 Mb region of chromosome 19. Bar indicates SEM. (C) Distribution of the crossover points at 9 Mb under inhibition of the BLM protein by ML216. We picked 126 colonies (three trials) and performed genotyping (S5 Fig). Ten $N^R P^R$-positive clones were isolated and analyzed using LOH panels and SNP arrays. Open squares indicate the heterozygous genotype. Closed black squares indicate the loss of heterozygosity (LOH) genotype. DSBs introduced by the CRISPR/Cas9 system are shown by scissors. (D) SNP array analysis of hiPSC-$BLM^{tet/tet}AAVS1^{cNP/+}$ ML216(+)9M-CRISPR (+) cells. (E) Comparison of crossover points introduced by suppression of $BLM$ transcripts (lanes 1 and 3) or inhibition of the BLM protein by ML216 (lane 2 and 4). DSBs were introduced at 14 Mb (lanes 1 and 2) or 19 Mb-I (lanes 3 and 4).ML216 was added at 0.5 day before transfection.

generated HLA-homozygous hiPSCs, and this strategy can be used for any human genomic region to induce homozygosity followed by functional genomics.

## Discussion

To establish HLA-homozygous hiPSCs, we used two genetic elements: the Tet-Off cassette to suppress $BLM$ expression and the cNP cassette for positive selection. However, for hiPSC-mediated therapeutics, it is preferable to avoid the use of exogenous genetic elements. We therefore also demonstrated successful substitution of the Tet-Off cassette with a small molecule BLM inhibitor, ML216. In the future, we hope to evaluate the effects of ML216 on the human genome by next generation sequencing, because $BLM$ suppression results in genome instability.

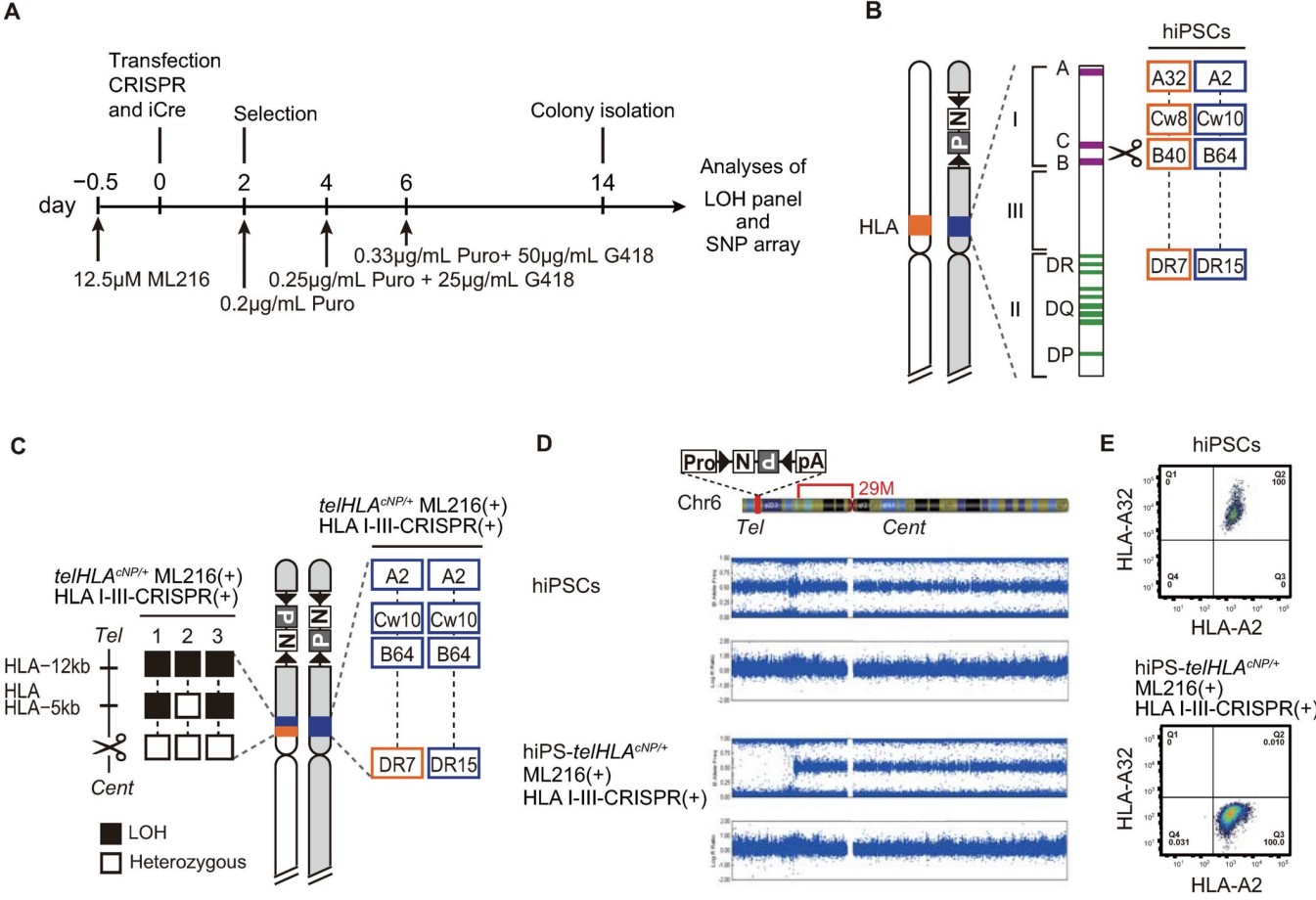

**Fig 4. Combination of BLM inhibitor ML216 and DSBs equidistant to the HLA class III and I genes in the short arm of chromosome 6 results in homozygosity of HLA class I.** (A) Procedure for the isolation of HLA-homozygous clones from hiPSC-*telHLA*$^{cNP/+}$ cells under the condition of BLM protein suppression. (B) Schematic representation of the short arm of chromosome 6. After targeting the double selection cassette (*telHLA*$^{cNP/+}$), an allele-specific DSB was introduced between HLA class I and III genes. The HLA-heterozygous genotypes in parental cells are shown in the right panel. (C) Schematic representation of the short arm of chromosome 6 after crossover. Patterns of heterozygous and LOH genotypes are shown in the left panel. HLA genotypes after crossover in clone 1 are shown in the right panel. (D) SNP array analysis of hiPSC-*telHLA*$^{cNP/+}$ML216(+)HLA I-III-CRISPR(+). (E) Flow cytometry profile of the surface expression of the HLA-A haplotype A2 and A32 in hiPSCs.

We cloned the cNP cassette into the piggyBac transposon vector that is removable by piggy-Bac transposase without any footprint after positive selection [14]. In addition, the cNP cassette was flanked by insulator sequences for its stable expression. By using the combination of the small molecule BLM inhibitor and piggyBac transposon vector, it may be possible to establish HLA-homozygous hiPSCs without any exogenous genetic elements.

Recently, Wang et al. reported Cas9-mediated chromosomal allelic exchange in mouse somatic tissue [15]. This novel technology resulted in the repair of compound heterozygous recessive mutations. However, unlike in our study, allelic exchange was expected to occur during the 2N phase, but not the 4N phase, so that any genomic region would not be homozygous.

HLA complex genes are located within the 6p21.3 region of the short arm of chromosome 6 and comprise more than 220 genes. Most of these genes are involved in immune functions. Because HLA genes are highly polymorphic, most individuals have two HLA haplotypes, i.e., the HLA-heterozygous state. When established hiPSCs or their differentiated cells, which are

most likely HLA heterozygous, are used for therapeutics as donor cells, the recipient must have the same combination of HLA haplotypes. However, HLA matching between a donor and recipient is very rare. Therefore, a bank of HLA-homozygous hiPSCs has been generated [16]. For the HLA-homozygous bank, HLA-A, HLA-B, and HLA-DR were selected for matching, but not HLA-C, HLA-DQ, or the minor histocompatibility antigens [16]. In contrast, our strategy is able to induce HLA homozygosity across the entire region of the HLA locus. Therefore, using this strategy, one could easily choose a much more compatible combination between the donor of HLA-homozygous hiPSC-derived differentiated cells and the recipient [17]. Although careful examination of HLA-homozygous hiPSCs is needed for cancer predisposition and effectiveness of differentiation, they may facilitate hiPSC-based transplantation related to the immunological differences between donors and hosts.

Other approaches to reduce the immunogenicity of hiPSCs and their derivatives have included the generation of haplotype-specific HLA antigens or the complete deletion of HLA antigens [11] [18] with the latter being accomplished by deletion of the HLA gene itself or *B2M* gene targeting [19]. However, it has been shown that cells lacking HLA-C antigens are susceptible to attack by natural killer cells [19] [20], posing a possible shortcoming. More recently, Xu et al. reported that HLA-C-retaining hiPSCs with HLA-A and -B pseudo-homozygosity and HLA-A and -B knockout generated by CRISPR/Cas9 showed enhanced immune compatibility [21]. hiPSCs produced using this new method retained either one copy or no copies of HLA-A and -B. However, our strategy allows retention of two homozygous copies of all the HLA antigens, which is arguably a safer approach than deleting entire HLA antigens.

In conclusion, the method described in this study offers a novel genetic tool for human biology.

## Supporting information

**S1 Fig. Removal of the selection cassette.** (A) Schematic representation of the BLM locus before and after removal of the selection cassette. Primers hBLM-endF and hBLM-endR were used to detect the removal. (B) Analyses of colonies after Flpo transfection by PCR with primers hBLM-endF and hBLM-endR. The selection cassette was deleted in clones #1 and #2, but not in clones #3–#6. M indicates the marker.
(PDF)

**S2 Fig. Comparison of BLM transcripts between parental hiPSCs and hiPSC-$BLM^{tet/tet}$ by RNA seq analyses.** FPKM derived from hiPSC-$BLM^{tet/tet}$ was divided by that from hiPSCs.
(PDF)

**S3 Fig. Measurement of *BLM* transcripts after 8 days of dox treatment.** The method to detect BLM transcripts was the same as that in Fig 1D.
(PDF)

**S4 Fig. Efficiency of iCre-mediated cassette inversion.** hiPSC-$BLM^{tet/tet}AAVS^{cNP/+}$cells ($5 \times 10^5$) were transfected with EF1$\alpha$-iCre or pBluescript (pBS), followed by selection with either 100 µg/mL G418, 0.75 µg/mL puromycin, or both. The efficiency of Cre-mediated cassette inversion was estimated at approximately 10 days after selection. Surviving colonies after the selections were fixed and stained with Giemsa stain. They are shown with blue dots.
(PDF)

**S5 Fig. Detection of positive clones bearing the crossover by genotyping of the AAVS1 locus in chromosome 19.** (A) Candidate clones were analyzed by competitive PCR using primers AAVS1-C-F, AAVS1-C-R, and Puro-L2 (S1 Table). hiPSC colonies after selection

were picked up and applied to PCR analysis or culture in 96-well plates. (B) Wildtype, neomycin resistance, and puromycin resistance alleles are represented by bands of 275, 407, and 1435 bp, respectively. The band derived from the wildtype allele is indicated by an asterisk. The positive pattern losing the band derived from the wildtype allele is shown in lane 3. M indicates 100 bp markers.
(PDF)

**S6 Fig. Distribution of crossover points in chromosome 19.** (A) Analyses of crossover points in chromosome 19 (9Mb, 14Mb, and 19Mb). (B) LOH panel of crossover points at 14 Mb. (C) LOH panel of crossover points at 19 Mb-I and 19 Mb-II.
(PDF)

**S7 Fig. SNP array analyses of the whole genome.** SNP array patterns were the same in hiPSCs and two clones [hiPSC-$BLM^{tet/tet}AAVS^{cNP/+}$DOX(+)9M-CRISPR(+) and hiPSC-$BLM^{tet/tet}AAVS^{cNP/+}$ML216(+)9M-CRISPR(+)] after crossovers except in predicted sites (red box). hiPSCs had abnormal chromosomes 3 and 4 as indicated by a black box. These abnormalities might have occurred during hiPSC generation, because parental fibroblasts did not contain these abnormalities.
(PDF)

**S8 Fig. Quantification of pluripotency marker gene expression.** FPKM (fragments per kilobase of exon per million mapped reads) was calculated from total RNA of parental fibroblasts (1), hiPSC-$telHLA^{cNP/+}$ (2), and hiPSC-$telHLA^{cNP/+}$ML216(+)HLA I-III-CRISPR(+) (3). (n = 3, error bars show SEM).
(PDF)

**S9 Fig. Flow cytometry profile of the surface expression of HLA-A haplotype A2 and A32 in hiPSCs-$telHLA^{cNP/+}$.**
(PDF)

**S1 Table. Primers used for vector construction and PCR genotyping.**
(PDF)

**S2 Table. Sequences of TALENs.**
(PDF)

**S3 Table. CRISPR information.**
(PDF)

**S4 Table. Primers used for SNP typing.**
(PDF)

## Acknowledgments

We thank Mitchell Arico and Natasha Beeton-Kempen, Ph.D., from Edanz Group (www.edanzediting.com/ac) for editing a draft of this manuscript.

## Author Contributions

**Conceptualization:** Junji Takeda.

**Formal analysis:** Yasuhide Yoshimura, Ayako Yamanishi, Tomo Kamitani.

**Funding acquisition:** Junji Takeda.

**Investigation:** Yasuhide Yoshimura.

**Resources:** Jin-Soo Kim.

**Writing – original draft:** Junji Takeda.

**Writing – review & editing:** Yasuhide Yoshimura, Ayako Yamanishi, Junji Takeda.

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
