## [Decision Letter · Decision Letter 0]

29 Aug 2019

PONE-D-19-19663

Generation of targeted homozygosity in the genome of human induced pluripotent stem cells

PLOS ONE

Dear Dr. Takeda,

Thank you for submitting your manuscript to PLOS ONE. After careful consideration, we feel that it has merit but does not fully meet PLOS ONE’s publication criteria as it currently stands. Therefore, we invite you to submit a revised version of the manuscript that addresses the points raised during the review process.

We would appreciate receiving your revised manuscript by Oct 13 2019 11:59PM. To enhance the reproducibility of your results, we recommend that if applicable you deposit your laboratory protocols in protocols.io, where a protocol can be assigned its own identifier (DOI) such that it can be cited independently in the future. For instructions see: http://journals.plos.org/plosone/s/submission-guidelines#loc-laboratory-protocols

We look forward to receiving your revised manuscript.

Kind regards,

Michael Kyba

Academic Editor

PLOS ONE

Journal Requirements:

Reviewers' comments:

Reviewer's Responses to Questions

**Comments to the Author**

1. Is the manuscript technically sound, and do the data support the conclusions?

Reviewer #1: Yes

Reviewer #2: Partly

2. Has the statistical analysis been performed appropriately and rigorously? 

Reviewer #1: Yes

Reviewer #2: Yes

3. Have the authors made all data underlying the findings in their manuscript fully available?

Reviewer #1: Yes

Reviewer #2: Yes

4. Is the manuscript presented in an intelligible fashion and written in standard English?

Reviewer #1: Yes

Reviewer #2: Yes

5. Review Comments to the Author

Reviewer #1: Yoshimura et al. have developed a new method to generate loss of heterozygosity (LOH) mutants in human induced pluripotent stems cells (hiPSCs).

This method is based on three components:

-Genetic or pharmacological inhibition of the Bloom helicase, which is known to result in hyper-recombination.

-CRISPR/Cas9 to cleave the genome at selected regions in an allele specific manner.

-A cassette for positive selection of LOH clones. As the Piggy bac system is used, these genetic manipulations leave no trace in the manipulated genome.

This ingenious strategy leads to a relatively high rate of LOH hiPSC clones. Authors have first validated their strategy on chromosome 19. Then, they managed to obtain LOH clones for some HLA genes on the short arm of chromosome 6. This is promising to increase histocompatibility, an important hurdle for hiPSCs-based cell therapy.

The objective of the study is clearly stated and the methods are scientifically sound.

The manuscript is well written. Overall, this is a very interesting study and I have only minor suggestions.

1.It would be interesting to analyse more independent clones (only 2 clones were analysed) for experiments on HLA (Figure 4).

2.It would have been interesting to measure whether pluripotency markers (POU5F1, NANOG, alkaline phosphatase activity…) are affected in manipulated cells.

3.If I am right, the double selection requires an inversion of the selection cassette and the use of a CRE. You mention that you used an iCRE but I have not understood well what drives the expression of the CRE and how efficient it is. Could you please comment on that and be more explicit?

4.Why starting the puro treatment before G418?

5.Even if out of the scope of this study it will be interesting to know whether LOH occurs in other unwanted DNA regions. The selection cassette permits to select for positive clones but this does not mean that other regions are not affected due to over recombination?

6.Line 443. It is mentioned that DSBs were either induced by CRIPSR/Cas9 or by TALEN. I thought that authors had only used CRIPSR/Cas9?

7.In fig.2C, clone 25, which behaves very differently from all other clones, is marked by an * but this particular clone is not commented.

8.Lines456. There are legends for panels E and F but these panels are missing in the figure.

9.Line 461. Change ‘suppression’ for inhibition as ML216 inhibits DNA binding and blocks BLM helicase activity, protein suppression let think that the protein is destroyed.

Reviewer #2: In this manuscript, Yoshimura and collaborators established a system to induce chromosomal crossover through suppression of BLM, by means of a Tet-off regulatory system or treatment with the small molecule inhibitor ML216, along with allele-specific double strand breaks induced by CRISPR/Cas9 in human pluripotent stem cells. Clones in which crossover was successfully achieved were selected through a conditionally convertible selection cassette. The authors then used this elegant approach to target chromosome 19 and 6, which resulted in homozygosity of the targeted regions as shown by SNP array analysis. Lastly, the authors showed a potential application of this approach as they induced homozygosity of HLA class I genes, which was evidenced by FACS analysis.

Although the ultimate goal of inducing crossover and selecting positive clones was achieved, as demonstrated by SNP analyses, there are major concerns in this work (see detailed description below). In this reviewer’s opinion, the most critical concern is the lack of controls and omission of experimental results. This does not allow the reader to perform a critical analysis of the work and leads to solely relying on the author’s description and schemes. Other concerns are related to manuscript structure, as the authors do not provide a clear rational on the gene editing strategy they followed and simply refer the reader to a previous publication (Yusa et al, Nature, 2004) which also lacks of detailed information or description of the system. This is important, as the reader should be able to understand at least the rational of the genetic engineering performed, especially if one is not familiar with previous publications on the approach used. An example is that the authors do not mention whether expression of tTA for BLM Tet-off system is under the native BLM promoter and the reader has to assume it. Another important aspect is that the authors do not acknowledge the limitations of their approach. For instance, as pointed out below, efficiency seems to be extremely low and there is no discussion about it. Moreover, authors should discuss that LOH has also been extensively associated with cancer. This is particularly relevant as the authors propose the use of their approach in regenerative medicine. Overall, this is an interesting study that could be highly improved if the authors include more experimental data and controls.

Specific observations:

1. The abstract does not provide a clear idea of the importance of the study. It starts suggesting that LOH can be used to identify genes associated with loss or gain of disease phenotype, but after describing the approach, it says that the system is useful in regenerative medicine. In this regard, it is clear that the approach is useful as a tool for genetic analysis, but in this reviewer’s opinion, the authors may want to reconsider suggesting its use in regenerative medicine as chromosomal crossover can cause genetic instability and it has been associated with cancer.

2. In the abstract, authors claim that ML216 treatment had similar crossover efficiency than Tet-Off-driven BLM suppression but there is no actual data to compare “efficiency”. Instead, both approaches successfully promoted crossover.

3. In author summary, authors once again suggest that chromosomal crossover might be useful for stem cell-based therapy. Consider revising this, otherwise authors should include in discussion association of crossovers with cancer or other potential limitations of the approach.

4. Introduction does not state clearly the relevance of the study or the utility of the approach. Instead, authors claim that it has been tested in other cell types/species but not in hiPSCs.

5. In line 211, authors mention that the CMV promoter was changed to hEF1a but the diagram on Fig. 1B still shows the CMV promoter.

5. It is not mentioned whether tTA is expressed under the native BLM promoter. If this is the case, it makes sense that BLM expression in parental and BLMtet/tet lines (No Dox) is similar. However, since this cassette is largely used in the study, it would be important to include this expression comparison.

7. Authors do not show experimental evidence on the removal of Pgk-Puro cassette used to select BLMtet/tet clones, this is critical as the authors rely on an additional positive Puro selection to identify crossover.

6. In line 91, authors describe that “Upon Cre expression, inversion of the cNP cassette occurred, resulting in loss of G418 resistance and gain of puromycin”, however, no experimental data are shown. This is a constant along the manuscript that needs to be addressed. This is important as the reader can get an idea of the efficiency of the system to induce inversion of the selection cassette.

7. In Fig. 1B, authors show a bar graph in which about 3 clones out of 1e6 cells were positive for the double (Puro/Neo) selection. Although this suggests that crossover was successfully achieved, the authors should comment on the low efficiency of the approach, which might be explained by all the variables in the system (e.g. Dox incubation, DSB in 4n phase etc).

8. There is no explanation on why the authors performed the experiments after 7 days with Dox (only two time points were analyzed for BLM expression, 5d and 7d).

9. In line 130, authors mention that when crossover occured centromeric to DSBs, the distance between the crossovers and DSBs was small. Then, a similar pattern is observed at 14Mb from centromere. However, according to Fig.2C, it seems that only one clone out of 25 resulted in crossover centromeric to the DSB (and one out of six in Fig. S2B). Are these numbers sufficient to conclude a “pattern” of centromeric vs telomeric crossover proximity?

10. Figure 2 legend has (E) and (F) but this is not shown in the figure.

11. The authors do not include any data on how effective the ML216 treatment is in their experimental setting.

12. Materials and methods say that ML216 was used 12h before transfection but figure legend (line 462) says that treatment began 24h before transfection.

13. Although a similar pattern of SNPs is observed between Tet-off and ML216, there is no data on the efficiency of ML216 treatment (how many positive clones?).

14. Fig 3C hiPS-BLMtet/tetAAVS1cNP/+ and Fig. 2D (same condition) are identical. If the same SNP analysis in Fig 2D was used as a reference for Fig 3C, it should be mentioned in the text to avoid misleading interpretation of intentional image duplication. Ideally, however, each assay (Fig 2D and 3C) should have been run in parallel with its own negative control.

15. In Fig. 3, ML216 treatment is used in cells that have the BLMtet/tet cassette. Again, showing that cassette insertion (No Dox) does not modify itself BLM expression would avoid concerns on the cassette background.

16. In Fig.4C, directionalities of the chromosome, DSB and crossover are confusing. The chromosome diagram shows that Tel is top and Cent is bottom, while this seems to be the opposite in the DSB diagram next to it?

17. Fig. 4E shows parental cells positive for HLA-A32 and HLA-A2. However, it would be more important to show the cells that were edited to include the selected cassette but did not have ML216 treatment (negative control). Furthermore, authors claim that these cells are HLA-homozygous hiPSCs without exogenous genetic elements (line 180) while no data show that the selection cassette was removed.

18. It would be useful to see whether the hiPSCs retained pluripotency after chromosomal crossover and all the genetic editing that they were subjected to.

19. Other experimental approaches such as FISH or Southern blot would provide more solid evidences on the crossover efficiency, cassette integration/removal, transposon removal etc.

6. PLOS authors have the option to publish the peer review history of their article (what does this mean?). If published, this will include your full peer review and any attached files.

Reviewer #1: No

Reviewer #2: No

---

## [Author Response · Author response to Decision Letter 0]

13 Oct 2019

Both reviewers provided us very nice comments. We revised our manuscript according to these comments. We believe that our new manuscript is much better than old one.

---

## [Decision Letter · Decision Letter 1]

29 Oct 2019

PONE-D-19-19663R1

Generation of targeted homozygosity in the genome of human induced pluripotent stem cells

PLOS ONE

Dear Dr. Takeda,

Thank you for submitting your manuscript to PLOS ONE. After careful consideration, we feel that it has merit but does not fully meet PLOS ONE’s publication criteria as it currently stands. Therefore, we invite you to submit a revised version of the manuscript that addresses the points raised during the review process.

We would appreciate receiving your revised manuscript by Dec 13 2019 11:59PM. To enhance the reproducibility of your results, we recommend that if applicable you deposit your laboratory protocols in protocols.io, where a protocol can be assigned its own identifier (DOI) such that it can be cited independently in the future. For instructions see: http://journals.plos.org/plosone/s/submission-guidelines#loc-laboratory-protocols

We look forward to receiving your revised manuscript.

Kind regards,

Michael Kyba

Academic Editor

PLOS ONE

Additional Editor Comments (if provided):

Thank you for the revised version. Please consider it to be accepted in principle.

However, Reviewer 2 points out several minor editorial issues. Please submit a revised final version incorporating those minor edits.

Reviewers' comments:

Reviewer's Responses to Questions

**Comments to the Author**

1. If the authors have adequately addressed your comments raised in a previous round of review and you feel that this manuscript is now acceptable for publication, you may indicate that here to bypass the “Comments to the Author” section, enter your conflict of interest statement in the “Confidential to Editor” section, and submit your "Accept" recommendation.

Reviewer #1: All comments have been addressed

Reviewer #2: (No Response)

2. Is the manuscript technically sound, and do the data support the conclusions?

Reviewer #1: Yes

Reviewer #2: Yes

3. Has the statistical analysis been performed appropriately and rigorously? 

Reviewer #1: Yes

Reviewer #2: Yes

4. Have the authors made all data underlying the findings in their manuscript fully available?

Reviewer #1: Yes

Reviewer #2: Yes

5. Is the manuscript presented in an intelligible fashion and written in standard English?

Reviewer #1: Yes

Reviewer #2: Yes

6. Review Comments to the Author

Reviewer #1: Authors have adequately addressed my comments in their revised manuscript.

The manuscript is technically sound, and the data support the conclusions.

SNP array datasets are available in GEO.

The manuscript is very well written and intelligible.

Reviewer #2: Yoshimura et al have submitted a modified version of the manuscript entitled “Generation of targeted homozygosity in the genome of human induced pluripotent stem cells”. In this new version, the authors have addressed most of the reviewer’s observations and concerns. Importantly, in this new submission the authors have included data that better support their statements. This also facilitates the analysis and understanding of the experiments performed. There are still a few observations on the current version that the authors may consider revising (see below), but overall, in this reviewer’s opinion, the manuscript is now suitable for publication.

1. Supplementary Figure 4 legend does not explain what the blue staining means, which is essential to understand the iCre-mediated cassette inversion selection.

2. In line 88, authors claim that efficiency of inversion was 20%-30% but it is not clear what they measured to obtain this number.

3. In line 134, revise “we established a two more DSBs” for “we established two more DSBs”.

4. In S6 Fig A-C consider changing M for Mb to be consistent with the text (eg. 9Mb, 14Mb, 19Mb-I)

5. In line 157, the authors start describing the generation of HLA-homozygous iPSC. This seems out of context since the rational of doing it is not mention in the introduction nor in the results section. Maybe a few sentences describing why the authors were interested on achieving HLA homozygosity in hiPSCs would make it easier for the reader to understand this section.

7. PLOS authors have the option to publish the peer review history of their article (what does this mean?). If published, this will include your full peer review and any attached files.

Reviewer #1: No

Reviewer #2: No

---

## [Author Response · Author response to Decision Letter 1]

9 Nov 2019

The reviewer's comments (especially #2) are very nice and constructive. I was very impressed. 

I will submit manuscript again in future.

---

## [Editor Report · Decision Letter 2]

12 Nov 2019

Generation of targeted homozygosity in the genome of human induced pluripotent stem cells

PONE-D-19-19663R2

Dear Dr. Takeda,

We are pleased to inform you that your manuscript has been judged scientifically suitable for publication and will be formally accepted for publication once it complies with all outstanding technical requirements.

With kind regards,

Michael Kyba

Academic Editor

PLOS ONE
---

## [Editor Report · Acceptance letter]

21 Nov 2019

PONE-D-19-19663R2 

Generation of targeted homozygosity in the genome of human induced pluripotent stem cells 

Dear Dr. Takeda:

I am pleased to inform you that your manuscript has been deemed suitable for publication in PLOS ONE. Congratulations! Your manuscript is now with our production department. 

With kind regards,

on behalf of

Dr. Michael Kyba 

Academic Editor

PLOS ONE